# Photoelectric Target Detection Algorithm Based on NVIDIA Jeston Nano

**DOI:** 10.3390/s22187053

**Published:** 2022-09-17

**Authors:** Shicheng Zhang, Laixian Zhang, Huayan Sun, Huichao Guo

**Affiliations:** 1Graduate School, Space Engineering University, Beijing 101416, China; 2Department of Electronic and Optical Engineering, Space Engineering University, Beijing 101416, China

**Keywords:** photoelectric targets, lightweight network, threshold segmentation, knowledge distillation, TensorRT acceleration

## Abstract

This paper proposes a photoelectric target detection algorithm for NVIDIA Jeston Nano embedded devices, exploiting the characteristics of active and passive differential images of lasers after denoising. An adaptive threshold segmentation method was developed based on the statistical characteristics of photoelectric target echo light intensity, which effectively improves detection of the target area. The proposed method’s effectiveness is compared and analyzed against a typical lightweight network that was knowledge-distilled by ResNet18 on target region detection tasks. Furthermore, TensorRT technology was applied to accelerate inference and deploy on hardware platforms the lightweight network Shuffv2_x0_5. The experimental results demonstrate that the developed method’s accuracy rate reaches 97.15%, the false alarm rate is 4.87%, and the detection rate can reach 29 frames per second for an image resolution of 640 × 480 pixels.

## 1. Introduction

Pinhole cameras, miniature cameras, and other photoelectric equipment pose a severe information leakage risk that may impact society and business in bringing substantial property losses and severe threats to personal safety. This can be addressed by using the cat-eye effect of cameras [1] for detection by using an incident-light irradiation camera that traces back the original camera light-path and the reflected light intensity of the diffuse reflected light, which has 2–4 orders of magnitude higher intensity. The spot generated on the photoelectric sensor or slicing board of the focal plane inside the camera due to the reflection of the cat-eye effect is called the cat-eye target. This method is independent of the camera’s operation and the transmission of wireless signals and only depends on the camera’s physical characteristics. Thus, using laser-active imaging [2] is one of the most effective ways to detect conventional cameras.

In the early years of the discipline, detection relied on the photoelectric target’s optical characteristics, primarily including the target’s light cross-sectional characteristics [3], the optical components’ characteristics [4], and the polarization characteristics [5]. Nevertheless, their detection rate is still low. The great success of AlexNet [6], VGGNet [7], and ResNet [8] since 2010 has set off an upsurge in convolutional neural network research. The convolutional neural network [9] has a multi-level feature extraction ability, widely used in computer vision and natural language processing. Some scholars suggest using deep learning to extract feature information [10] of optoelectronic targets. For instance, Ke [11] designed a fully automatic camera detection and recognition system, which combines machine learning and neural network methods to identify surveillance camera equipment effectively. Liu [12] introduced a photoelectric target recognition algorithm and a detection system based on convolutional neural networks to detect indoor micro-cameras using classified networks. Huang [13] used the improved YOLOv3 model to identify micro-cameras in a single frame. Nevertheless, the above methods are limited by the high computing power of the hardware affecting detection speed and performance, resulting in the detection equipment having an excessive volume and thus poor portability limiting its applicability.

Spurred by the deficiencies mentioned above, this paper develops a photoelectric target detection algorithm for the NVIDIA Jeston Nano, a small, embedded device with certain computing power limitations. In order to ensure the good real-time performance of the detection process, an adaptive threshold segmentation method based on the statistical characteristics of the photoelectric target echo light intensity is proposed during image preprocessing [14]. This strategy enhances the algorithm’s performance on the embedded device by reducing complexity. Considering deep learning [15], the prediction accuracy of lightweight networks is improved by knowledge distillation [16], and the model inference speed is greatly improved using TensorRT technology [17]. Hence, the entire algorithm process is suitable for embedded devices with small computing power [18], affording real-time detection of photoelectric targets in real-world scenarios.

The remainder of this article is organized as follows. Section 2 describes the equipment and processes, datasets, and experimental environments for data collection. Section 3 proposes an adaptive threshold segmentation method based on the statistical characteristics of photoelectric target echo light intensity. Additionally, it introduces image preprocessing, target region discrimination, and model acceleration. Section 4 analyzes the experimental results and verifies the effectiveness and real-time nature of the algorithm’s detection of photoelectric targets. Finally, Section 5 concludes this work.

## 2. Materials and Equipment

### 2.1. Data Acquisition Equipment and Processes

Imaging methods can be divided into active and passive imaging, depending on whether or not they use active light sources to irradiate the target and its background. Active imaging exploits the illumination light source (typically a laser) to irradiate the target and its background, with the advantage of enhancing the target information and not being susceptible to environmental factors. Passive imaging utilizes the natural light reflecting on the target and background, affording rich acquisition information and easy acquisition.

Experimentally, a photoelectric target laser active imaging detection system was built to acquire active and passive images (Figure 1). The developed setup included a laser with a wavelength of 532 nm and an industrial CCD camera as the detector, acquiring images of 640 × 489 pixels. The data collected by the photoelectric target involved a conventional camera with a diameter of 20 mm, and the shooting distance was 60–80 m.

During image acquisition, the laser and detector were controlled by synchronous triggers. The detector acquired images every time the laser changed its operating state, and the detector’s operating frequency was twice that of the laser, ensuring smooth acquisition of active and passive images.

### 2.2. Dataset

The datasets in this experiment include exclusively authentic shots. During the experiment, highly reflective substances such as folded tin foil, plastic bottles, and metals were added to simulate false targets and increase the background complexity. In total, 2540 active and 2540 passive images were collected with the setup illustrated in Figure 1. Based on the characteristics of the active image enhancement target information, the data set was collected and labeled in the active image (Figure 2). The camera lens’ reflected flare (the red area in the image) is the dataset’s true target. Specks of light reflected (green areas in the image) by targets, such as metal near the window and stones on the road, serve as false targets. The 2540 active images containing photoelectric targets from outdoor scenes were cropped, and 6083 dataset true target images and 6083 dataset false target images were created. After cropping, each image was of size 20 × 20 pixels, which was increased to 64 × 64 when entering the network.

### 2.3. Experimental Environment

The hardware employed to train the deep learning model and perform knowledge distillation was Inter (R) Core (TM) i7-9700F CPU@3.00GHz, GTX TITAN XP with 12 GB GPU memory, and Windows 10 operating system with 64 GB system memory. The testing environment and TensorRT acceleration utilized an NVIDIA Jeston Nano 4 GB embedded development board, with a quad-core ARM Cortex-A57 MPCore processor as CPU and a NVIDIA Maxwell architecture as GPU, equipped with 128 NVIDIA CUDA^®^ cores, providing 472 GFlops of computing performance.

## 3. Methods

The developed algorithm flowchart is illustrated in Figure 3. Specifically, the acquired active and passive images undergo differential processing and thus are converted to differential images. In the latter images, the proposed adaptive threshold segmentation method is applied based on the statistical characteristics of the echo light intensity of the photoelectric target within the differential image. Using the above method and the shape measurement criterion of the photoelectric target, the target within the suspected target area is judged by the validity, and the screened target area was mapped to the active image as a candidate region for retention. The candidate region in the active image is input to the convolutional neural network for discrimination to obtain accurate photoelectric target information. Finally, the discriminant result is marked as the true target on the active image. However, utilizing convolutional neural networks directly on embedded devices is not processing-efficient, as it does not meet the real-time requirements for hardware platform deployment. Therefore, the network model must be accelerated, which is accomplished by developing a lightweight network after applying knowledge distillation and TensorRT technology. This strategy accelerates the inference on the lightweight network and affords its deployment on the hardware platform while meeting the real-time detection requirements.

This chapter contains three parts: image preprocessing, candidate region discrimination, and model acceleration. The image preprocessing part proposes an adaptive threshold segmentation method based on the statistical characteristics of the cat-eye target echo light intensity. This segmentation scheme and the cat-eye target shape measurement criterion are used to segment the target and the background while eliminating image noise. This strategy eliminates pseudo-targets and provides the basis for subsequent target identification works. The candidate region discrimination part accurately classifies the candidate region obtained from the convolutional neural network, and the model acceleration part solves the problem of the algorithms not meeting the real-time deployment on NVIDIA Jeston Nano, by utilizing knowledge distillation and TensorRT acceleration.

### 3.1. Image Preprocessing

Initially, the acquired active and passive images undergo morphological processing. Since the photoelectric target appears as a circle in the image and occupies a small number of pixels, the Tophat transformation using the circle as a structural element can suppress the background factor and enhance the outline of the small target effect. Given that the active image can enhance the target’s pixel intensity, the passive image has rich information acquisition, and the image difference operation is used to eliminate the influence of background factors on target recognition.
(1)$$\begin{array}{c} I_{sub}=I_{a}-\frac{\left[I_{a}+\left(N-1\right)I_{p}\right]}{N} \end{array}$$
where $I_{a}$ and $I_{p}$ are the active and passive images, respectively, and N is the total pixel value of the grayscale image. Figure 4 depicts a set of images and their image differentiation output.

The photoelectric target irradiates the photoelectric device with an active light source, and due to the principle of reversibility of the optical path, the incident light returns along the original optical path at the focal plane within the photoelectric device. This property of the optoelectronic devices is also known as the “cat’s eye effect”. As illustrated in Figure 4a, other spurious targets are also detected when the camera is detected. In that case, the echo power of the photoelectric target and the diffuse reflection is theoretically analyzed. The echo spot area received by the detector can be expressed as
(2)$$S=\pi {\left(\frac{R\theta^{\prime }}{2}\right)}^{2}$$
where $\theta^{\prime }$ is the reflex angle of the photoelectric target. The photoelectric target echo power P received by the light-sensitive surface at the camera’s focal plane has an effective area $A^{r}$.
(3)$$P=P_{t}\frac{4\tau^{2}\tau_{t}\tau_{s}^{2}\tau_{r}A_{s}A_{r}\rho }{\pi \theta^{2}R^{2}S}$$
where $P_{t}$ is the emitted laser power, τ is atmospheric transmittance, $\tau_{t}$ denotes the transmittance of the emission optics systems, $\tau_{s}$ is the photoelectric target optoelectronic device transmittance, and $\tau_{r}$ is the receiving optical system. $A_{s}$ is the photoelectric target that has a limited aperture receiving area transmittance, $A_{r}$ denotes the echo detector receiving area, θ is the beam dispersion angle of the emitted laser, and R is detection range. The diffuse reflection for a small target echo power $P_{rs}$ is given by
(4)$$P_{rs}=P_{t}\frac{2\tau^{2}\tau_{t}\tau_{r}A_{r}\rho_{rs}A_{rs}\cos\varphi }{\pi \theta^{2}R^{4}}$$
where $\rho_{rs}$ is the reflectivity of a diffused small target, $A_{rs}$ denotes the effective cross-sectional area of a diffuse reflection small target, and φ is the diffuse reflection of the laser incidence’s angle at a small target. Comparing Equation (3) with Equation (4) yields the photoelectric target echo power and the diffuse small target echo power:(5)$$\frac{P}{P_{rs}}=\frac{8\tau_{s}^{2}A_{s}\rho }{\theta^{\prime 2}A_{rs}\rho_{rs}\cos\varphi }$$
where $\theta^{\prime }$ is the echo reflection angle. In practical application scenarios, the photosensitive surface reflectance ρ of the photoelectric target is 0.3, the reflectivity of most diffuse targets is between 0.2–0.5, and the light wave transmittance $\tau_{s}$ of the photoelectric system is above 0.95. Under the condition that the target’s effective area does not significantly vary and the light is incident, the photoelectric target echo power and the diffuse small target echo power are approximated as follows:(6)$$\frac{P}{P_{rs}}=\frac{8}{\theta^{\prime 2}}$$

When comparing the photoelectric target echo power with the diffuse small target echo power, their ratio is constant, unaffected by the operating distance changes. Therefore, the echo power intensity of the photoelectric target is greater than the echo power intensity of the small target’s diffuse reflection. This characteristic is exploited as the theoretical basis for distinguishing true and false targets.

However, minor noise exists in the difference image, primarily from the strong noise and edge features in the active image’s background, Therefore, the difference image is further processed to solve this problem. The camera generates a strong echo signal due to the “cat’s eye” originating from the laser’s active detection, which has a mountain peak shape, where the peak pixel value is relatively the average in the image gray value transformation. In other words, the low pixel value of the background wraps the characteristics of the target’s high pixel values. Although the high-brightness pseudo-target area has a mountain peak shape in the image gray value transformation, its peak value pixels are not evenly distributed but present an irregular and uneven shape (Figure 5).

The real target represented in Figure 5a has a single peak identified with a smooth mountain peak straight up and down the mountain. The detector’s light saturation causes the smoothness of its summit. Opposing, the pseudo-target in Figure 5b has a continuous multi-peak feature. The area around the peak fluctuates more violently than the true target due to the changing pixel values between the pseudo-target and the surrounding pixels affected by diffuse reflection. These two characteristics are used to judge a potential target.

The experiment was mainly oriented to the NVIDIA Jetson Nano hardware platform. The traditional filtering and Fourier transforms [19] impose a substantial computational burden, not guaranteeing an acceptable real-time performance to small computing power equipment. Therefore, the threshold segmentation method is selected, which has minimum computational requirements. Despite that, the photoelectric target is not effectively segmented, as according to the statistical analysis of the differential image data, the pixel proportion of the photoelectric target is small. As illustrated in Figure 6, most of the pixels in the differential image are distributed in the low-brightness area. Thus, by analyzing the histogram of characteristics, an adaptive threshold segmentation method is proposed based on the statistical characteristics of the echo light intensity of the optoelectronic target.

Figure 6 depicts the different images with grayscale histograms of 640 × 480 pixels in the outdoor environment. Figure 6a contains many low pixel regions, and the corresponding histogram distribution is not uniform, indicating that the difference between the background and the grayscale targe values is too large. We divide the difference image into low and high luminance regions to solve this problem. The former region does not affect the threshold segmentation, as only the high luminance region is considered during threshold segmentation. Dividing the gray value between the low and the high luminance region is particularly important. Consider Figure 6 as an example: in the differential image (a), the maximum gray value of the Target A region (pseudo-target area), the Target B (true-target area) region are 198 and 199, respectively. Considering the actual situation, the high brightness reflector gray value is greater than the photoelectric target gray value. Thus, we experimentally found that splitting the high and low brightness region gray values based on the entire image’s highest 1% gray value of the first-pixel point is the most appropriate choice. This value was then used as a threshold to segment the high brightness region:(7)$$\begin{array}{c} T=\frac{m\;*\;I_{max}+I_{th}}{m+1} \end{array}$$
(8)$$\begin{array}{c} m=\frac{I_{max}-I_{th}}{k} \end{array}$$
where $I_{max}$ is the largest pixel value point in the image, k is a constant, m represents the maximum gray value in the image, and $I_{th}$ is the difference degree of the background of the high-intensity region. When $I_{th}$ is small, the high-intensity area has fewer false targets, and m can be approximated as the difference degree between the photoelectric target and the background of the high-intensity area. The greater the difference degree between the photoelectric target and the high-intensity background while solving for the threshold T, the higher the weight of m, and the more that the threshold T tends to be the maximum gray value. The comparison after threshold segmentation is presented in Figure 6c.

To further improve screening accuracy, the shape characteristics of the photoelectric targets are analyzed, and the shapes of photoelectric targets in different cameras and sites in the dataset are counted. The dataset is 60–80 m apart in an outdoor environment using a 532 nm laser and CCD industrial camera to shoot a conventional camera with a diameter of 20 mm. The image size acquired is 640 × 480 pixels, the photoelectric target is a solid area on the image after threshold segmentation, and the pixel area is between 3 × 3 and 6 × 6. The optical aperture of the photoelectric equipment is mostly circular. Therefore, the shape of the photoelectric target in the image echo signal was considered as solid and approximately circular, as illustrated in Figure 7.

According to the shape characteristics analyzed in the above experiments, the final candidate region was removed from the deformed region with a length-to-width ratio greater than 2:1 and the spot region with an area of fewer than three pixels. The shape measurement before and after the comparison chart is depicted in Figure 8, where the real photoelectric targets are marked with red boxes, parameter A represents the area of pixels occupied, and parameter B represents the aspect ratio. The spot inside the white circle is deleted because it does not meet the shape measurement criteria. Only the spot inside the red circle is retained as a candidate area for the following process.

### 3.2. Candidate Region Discrimination

The entire preprocessing stage can simply and effectively remove a part of the false area. Considering the algorithm’s complexity, the proposed method is more friendly to low-computing devices. Since the photoelectric target occupies fewer pixels in the image, and its optical characteristics that can be extracted are limited, it is difficult to test the photoelectric target in all scenarios accurately. Therefore, a convolutional neural network is used to identify candidate regions above to improve the algorithm’s detection ability.

In-depth studying of the convolutional neural networks reveals that increasing the network’s depth introduces more nonlinear features, making the network difficult to train. Thus, He proposed the residual module contained in ResNet [8] to solve this problem. Therefore, ResNet is selected as the feature extraction network and the teacher network for the following knowledge distillation part (see the Experimental validation in Section 4.2). The following analysis and experimental verification are made based on the ResNet layers:
The large difference in the number of layers between the teacher and the student models in knowledge distillation has a negative effect on the distillation results [20]. Since the hardware platform has small computing power, the student model will be a lightweight network, so the teacher model can only be a low-level network, such as ResNet18, ResNet34, and ResNet50.Since the image pixel of this training set is small and the target feature information is relatively single, increasing the number of convolutional layers may discard part of the acquired low-level feature information. The prediction accuracy of ResNet18, ResNet34, and ResNet50 after training is 99.54%, 99.17%, and 98.72%, respectively.

By analyzing the experimental data and considering the relevant theory, ResNet18 was selected as the feature extraction network and teacher network of the subsequent knowledge distillation part. Through this network, the candidate region after image preprocessing can be discriminated, and if the discriminant result is the real target, the detection result can be obtained by marking the region on the original image. The candidate area may simply be discarded if the discriminant result is a false target.

### 3.3. Model Acceleration

Due to the hardware limitations of embedded devices, deploying the ResNet18 model directly on the device will lead to high execution times and, thus, poor real-time performance. Thus, the ResNet18 must undergo a knowledge distillation process, and TensorRT technology can be used to accelerate reasoning, effectively solving the above problems.

In 2014, Hinton [16] from Google LABS first proposed the idea of complete knowledge distillation and experimentally verified its feasibility and the effectiveness of convolutional neural network compression on MNIST data sets. In the probability distribution of the output of a well-trained network model, the probability of the error category is generally small, and its relative probability hides the characteristic information that only the real tags of 0 and 1 do not have. Therefore, the temperature coefficient T is added to the output layer of Softmax to smooth the probability distribution of the network’s output. The output is called soft target, which guides the students’ network training and the real label. The corresponding loss function $J_{KD}$ is obtained as follows, where Equation (10) is a supplementary description of Equation (9):(9)$$\begin{array}{c} J_{KD}=J_{CE}\left(y_{true},p\right)+\gamma T^{2}J_{CE}\left(\hat{p_{l}},\hat{q_{l}}\right) \end{array}$$
(10)$$\begin{array}{c} \hat{q_{l}}=\frac{exp\left(\frac{z_{i}}{T}\right)}{\sum_{j}exp\left(\frac{z_{i}}{T}\right)} \end{array}$$

Neural networks typically generate class probabilities using a softmax output layer, which normalizes, $z_{i}$ to probability $\hat{q_{l}}$. $J_{CE}\left(y_{true},p\right)$ represents the cross-entropy between the predicted output of the student network and the true label, and $J_{CE}\left(\hat{p_{l}},\hat{q_{l}}\right)$ is the cross-entropy between the predicted output after smoothing by the student network and the teacher network. Also, γ is a hyperparameter that adjusts the ratio of the two loss functions, and because the cross-entropy is smoothed by hyperparameter T, its gradient will become the original $1/T^{2}$ during back-propagation. To preserve the scale of its gradient consistent with the scale of the cross-entropy corresponding to the true label, it is necessary to multiply the smoothed cross-entropy by $T^{2}$.

TensorRT Quantization technology [21] is a high-performance neural network reasoning optimizer launched by NVIDIA that provides low-delay and high-throughput deployment reasoning for deep-learning applications. It can accelerate reasoning for super-large-scale data centers and embedded or autonomous-driving platforms. It supports deep learning for most frameworks for fast and efficient deployment reasoning. TensorRT relies on Computeunified Device Architecture (CUDA), including an industry-level library explicitly created for deep-learning model deployment. It optimizes trained neural network models, usually 32-bit or 16-bit data. According to the programmability of CUDA, TensorRT can optimize increasingly complex and diverse deep-learning neural networks.

## 4. Results

### 4.1. Analysis of Threshold Segmentation Results

The detection performance of the proposed threshold segmentation method was experimentally validated and challenged against other threshold segmentation methods. The corresponding results are illustrated in Figure 9.

Figure 9(a1,b1) illustrate the active and passive images of a single photoelectric target with false targets, while Figure 9(a2,b2) depict the active and passive images of multiple photoelectric targets in a simple environment. The real photoelectric targets are marked with red boxes in the active image. The experimental comparison reveals that the proposed method has more advantages than traditional methods for photoelectric target segmentation. In a simple environment, the segmentation effect is ideal, while when there are false targets such as high brightness reflectors, only a part of the false targets can be screened, and the remaining must be further discriminated.

Figure 9 reveals that the developed method has an appealing effect on suppressing background factors and enhances the contrast of photoelectric targets well, providing convenient conditions for the deep learning model to discriminate the target area. This is especially important as it compensates for the algorithmic complexity of embedded platforms with less computational power.

### 4.2. Analysis of Knowledge Distillation

This section conducts experimental verification and analysis of the teacher and student network selection to achieve a better knowledge-distillation effect. Regarding the teacher network selection, three classical neural networks (VGG16, AlexNet, and Resnet18) were selected to train the dataset. The top-1%, parameter amount, calculation amount, and average inference time of 30 inferences in NVIDIA Jeston Nano are reported in Table 1 (- indicates that the resources occupied by NVIDIA Jeston Nano are too large to run).

Table 1 shows that the prediction accuracy of all competitor teacher networks is almost the same, while only the computational burden and the parameter cardinality vary, resulting in different inference times on the NVIDIA Jeston Nano device. Since this cannot be a critical factor in selecting the teacher network, a comparative experiment of knowledge distillation between the teacher and student networks was conducted. The teacher network remained one of the above three classic neural networks, and the student network was one of the lightweight networks: Shuffv2 [22], Squeezent [23], GhostNet [24], and CondenseNetv2 [25]. The parameters of top-1%, parameter quantity, calculation amount, and average inference time of 30 inferences in NVIDIA Jeston Nano after knowledge distillation on the four lightweight networks are reported in Table 2.

Table 2 shows that the accuracy improvement of the lightweight network after knowledge distillation is more significant than Resnet18. Thus, ResNet18 was selected as the teacher network for knowledge distillation. Since the lightweight network model was to eventually be deployed on the NVIDIA Jeston Nano embedded development device, we also analyzed the process of selecting the lightweight network. To ensure that the algorithm is executed in real-time, the inference time should be considered first in the actual deployment. Thus, Squeezent1_1, Squeezent1_0, and Shuffv2_x0_5 have advantages over other lightweight networks. Therefore, only Squeezent1_0 and Shuffv2_x0_5 were selected based on the number of parameters and the computational load on the hardware platform. Also, the prediction accuracy had a more significant impact on the results. Thus, this index is also one of the factors to be considered. Shuffv2_x0_5 had a higher prediction accuracy of nearly 5% compared to Squeezent1_0 and had a better detection performance. Therefore, the Shuffv2_x0_5 lightweight network is more appropriate for deployment on the hardware platform. Figure 10 depicts the final network structure of the teacher and student models after a comparative analysis of the experimental data.

### 4.3. Analysis of the Impact of Preprocessing on the Algorithm

This section verifies the effectiveness of the adaptive threshold segmentation method on the entire algorithm process. The segmentation scheme is based on the statistical characteristics of the photoelectric target echo light intensity. This section compares the number of candidate regions extracted by the threshold segmentation method and analyzes the effectiveness of the threshold segmentation method on the entire algorithm process. Due to the poor effect of some traditional threshold methods (see Section 4.1), this section only compares the screening detection area considering the Otsu threshold segmentation method. The candidate areas of 100 groups of active and passive images collected from outdoor scenes were selected, including 123 real targets. The shape characteristics of the candidate areas were screened consistently during the experiment, and only the threshold segmentation method was compared. The corresponding experimental results are illustrated in Figure 11.

The red areas in Figure 11d,f represent the candidate region detected by the Otsu and the proposed threshold segmentation methods, respectively. The experimental comparison demonstrates that the developed threshold segmentation method significantly reduced the candidate region detected by the Otsu method and accurately contains the real target area. Especially for dynamic scenes, the suggested threshold segmentation method effectively filters out the peoples’ or targets’ relative motion trajectory, preserving the photoelectric target’s characteristic information.

In photoelectric target recognition, the detection rate $P_{d}$ and false alarm rate $P_{f}$ are important indicators to evaluate detection performance. The detection rate $P_{d}$ is given by
(11)$$P_{d}=\frac{N_{d}}{N_{t}}$$
where $N_{d}$ and $N_{t}$ are the number of real targets detected and existing in the image, respectively. The false alarm rate $P_{f}$ is given by
(12)$$P_{f}=\frac{N_{f}}{N_{r}}$$
where $N_{f}$ is the number of false targets detected in the image and $N_{r}$ is the number of targets detected in the image. The number of candidate regions generated by the threshold segmentation method affects the detection rate, false alarm rate, and inference time of the photoelectric target detection algorithm and will ultimately impact the cat’s eye detection performance.

The experimental results presented in Table 3 reveal that the Otsu threshold segmentation method produces many candidate regions, decreasing the algorithm detection rate and increasing the false alarm rate. More importantly, Otsu’s method significantly prolongs the algorithm’s inference time, which is not conducive to its real-time detection performance. In contrast, the suggested adaptive threshold segmentation method based on the statistical characteristics of photoelectric target echo light intensity achieves an appealing performance in detection rate, false alarm rate, and inference time, verifying the effectiveness of the threshold segmentation method for cat-eye target recognition and the importance of the entire algorithm process.

### 4.4. TensorRT Acceleration and Algorithm Detection Results

This paper uses TensorRT technology [26] to afford the detection process, achieving real-time performance on NVIDIA Jeston Nano. TensorRT is more suitable for hardware to accelerate the inference process. Furthermore, this paper completes the transformation of ONNX and constructs the acceleration engine (hereafter referred to as ShuffEng) based on knowledge distillation to obtain the optimal network model. Since the NVIDIA Jetson Nano platform only supports F32 precision, the engine adopts the full precision mode for inference. Figure 12 illustrates a comparative test of the forward inference between ShuffNet and ShuffEng accelerated by cudnn in GPU mode on NVIDIA Jetson Nano.

From the test set, 500 target candidate areas were selected and entered into ShuffNet and ShuffEng for discrimination. The discriminant results were evaluated using the detection rate $P_{d}$ and the false alarm rate $P_{f}$, as shown in Table 4. The weight precision of ShuffNet and ShuffEng is 32 bits. The above experimental results reveal that the detection results of ShuffNet and ShuffEng inference have not changed.

The subsequent trial is a comparative experiment on the inference time of ShuffNet and ShuffEng, involving an inference on 60 active and passive images with a resolution of 640 × 480 pixels on NVIDIA Jeston Nano. Since this method exploits the image’s preprocessing phase to determine the candidate region, the number of candidate regions will affect the inference time. In the process of selecting images, 30 images are single photoelectric target images, and 30 are multi-photoelectric target images. Figure 13 depicts the inference time of the entire algorithm under different conditions, revealing that the inference time of the entire algorithm process is relatively stable. Table 5 reports the average time of 30 inferences for ShuffNet and ShuffEng.

The experimental comparison highlights that ShuffEng has the same weight accuracy as ShuffNet but presents more advantages in inference. The ShuffEng generated in this paper achieves a 2.31 times acceleration ratio in single-optical target detection inference and 2.96 times acceleration ratio in multi-optical target detection inference, significantly improving the model’s inference speed on the embedded platform. Its running time can be controlled in the range of 0.03–0.05 s, and the detection rate can reach 29 frames per second, meeting the real-time processing requirement of at least 20 frames per second in the video detection process. Therefore, the proposed method has the potential to be used in practical applications.

## 5. Conclusions

This paper proposes and verifies a new method of photoelectric target detection based on NVIDIA Jeston Nano. The method mainly includes image preprocessing, knowledge distillation, TensorRT acceleration, and lightweight network target discrimination. The experimental results demonstrate that the developed adaptive threshold segmentation method based on the statistical characteristics of optical target light intensity improves detection of the target region more effectively than the traditional threshold segmentation methods. The layer-to-layer guidance of knowledge distillation is used to improve the accuracy of the lightweight network and TensorRT technology is used to realize accelerated reasoning on the NVIDIA Jeston Nano hardware platform. The proposed method has higher accuracy and lower requirements on hardware computing power and can quickly identify multiple photoelectric targets. The entire process has good real-time performance, making the lightweight network more valuable in deploying low computing power devices.

## Figures and Tables

**Figure 1 sensors-22-07053-f001:**
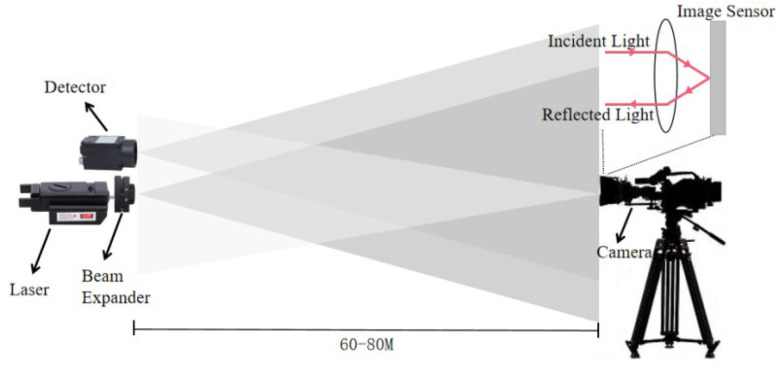
The data acquisition process of photoelectric target laser active imaging detection system.

**Figure 2 sensors-22-07053-f002:**
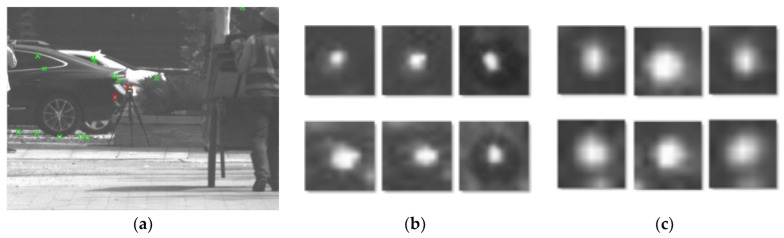
Sample images of a partial dataset. (**a**) Active image labeled with a true target in the dataset, (**b**) part of the true target dataset, and (**c**) part of the false target dataset.

**Figure 3 sensors-22-07053-f003:**
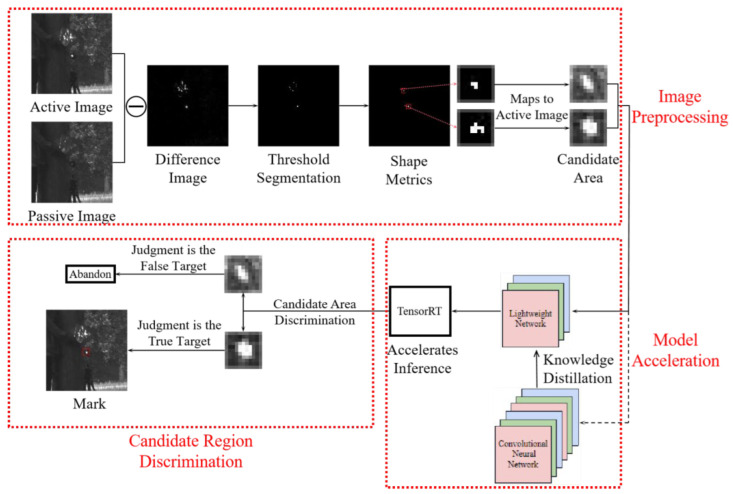
Algorithm flowchart of this article.

**Figure 4 sensors-22-07053-f004:**
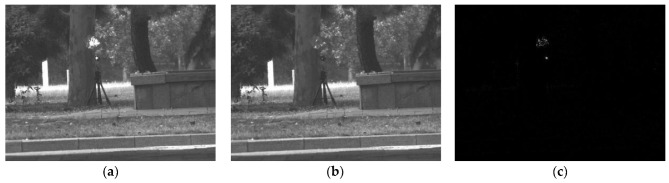
Image differentiation. (**a**) Active image, (**b**) passive image, and (**c**) differential image.

**Figure 5 sensors-22-07053-f005:**
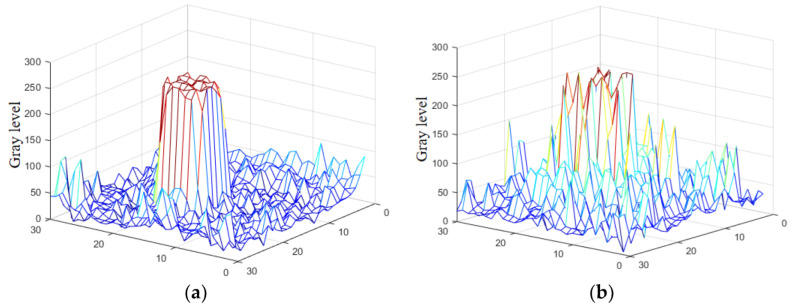
Three-dimensional view of the pixel values belonging to real and pseudo targets. (**a**) True target local pixel value and (**b**) partial pseudo-target local pixel value.

**Figure 6 sensors-22-07053-f006:**
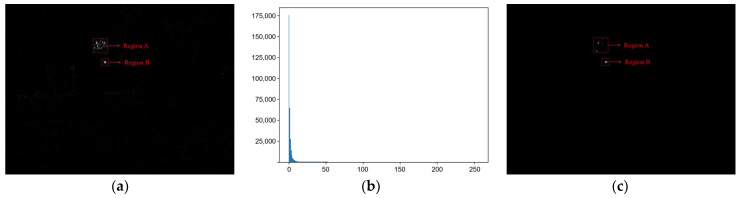
Histogram statistics in outdoor environment: (**a**) differential image containing the photoelectric target, (**b**) histogram of the differential image, and (**c**) image containing the segmentation of the photoelectric target threshold.

**Figure 7 sensors-22-07053-f007:**
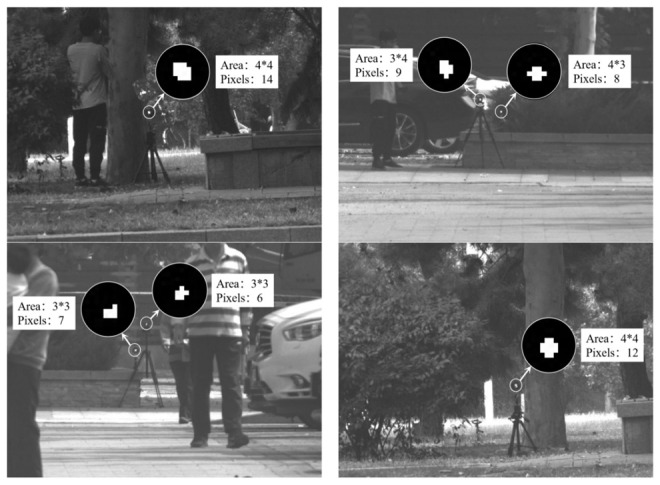
Photoelectric device echo shape.

**Figure 8 sensors-22-07053-f008:**
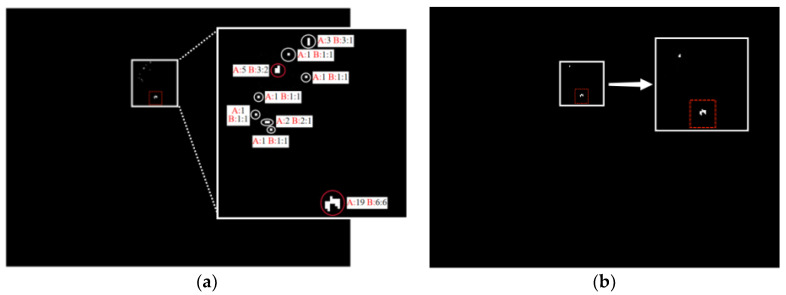
Comparison of before and after shape screening (**a**) image after threshold segmentation and (**b**) image after shape measurement.

**Figure 9 sensors-22-07053-f009:**
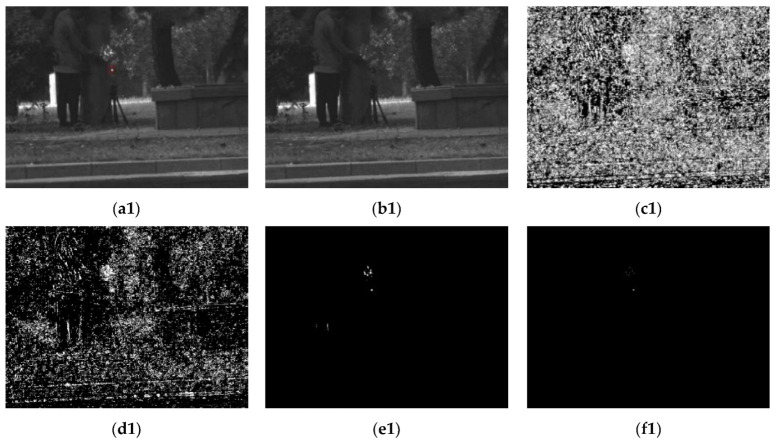
Comparison between the proposed and traditional methods: (**a1**,**a2**) active image, (**b1**,**b2**) passive image, (**c1**,**c2**) maximum entropy threshold segmentation, (**d1**,**d2**) iterative threshold segmentation, (**e1**,**e2**) Otsu-based threshold segmentation, (**f1**,**f2**) proposed threshold segmentation.

**Figure 10 sensors-22-07053-f010:**
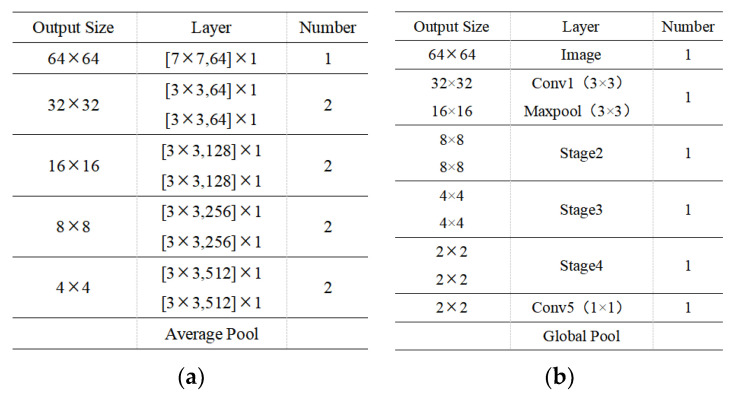
Network structure of the teacher and student models. (**a**) Resnet18 network structure (teacher model) and (**b**) Shuffv2_x0_5 network structure (student model).

**Figure 11 sensors-22-07053-f011:**
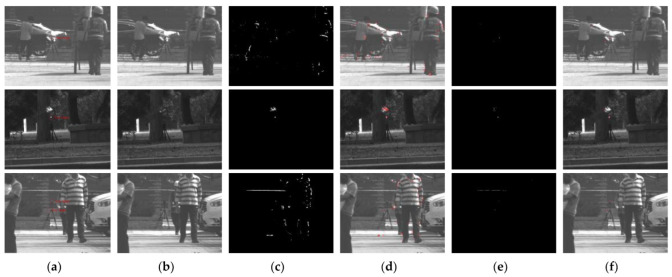
Comparison of the number of candidate regions produced by different threshold segmentation methods: (**a**) active image, (**b**) passive image, (**c**) Otsu-based segmentation, (**d**) candidate area segmentation based on the Otsu threshold, (**e**) proposed threshold segmentation effect, (**f**) candidate area segmentation based on the proposed threshold.

**Figure 12 sensors-22-07053-f012:**
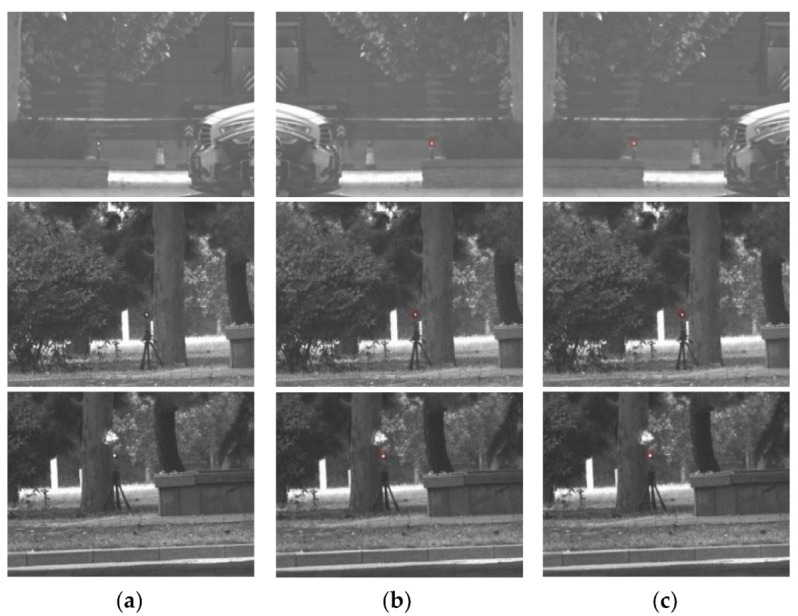
Forward reasoning comparison test: (**a**) active image, (**b**) inference result of ShuffNet, (**c**) inference result of ShuffEng.

**Figure 13 sensors-22-07053-f013:**
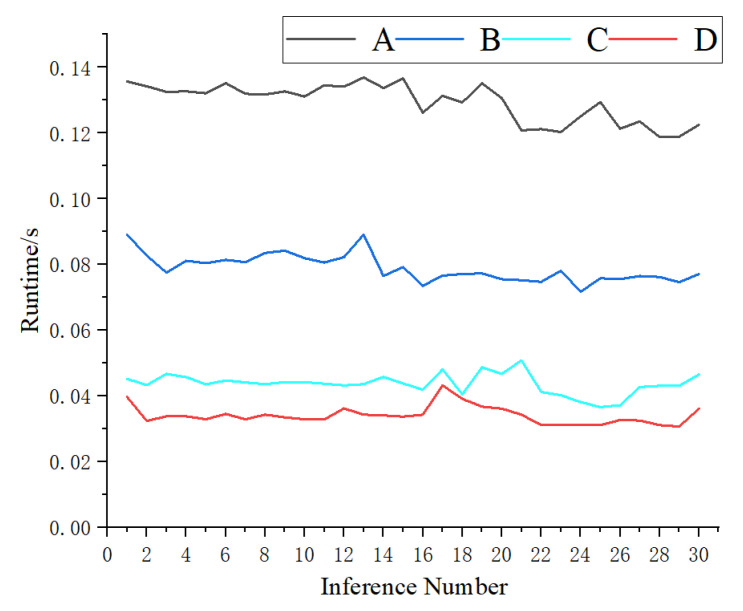
Comparison of running time between single-target area and multi-target area. (**A**) ShuNet inference for multi-target regions, (**B**) ShuNet inference for single-target regions, (**C**) ShuEng inference for multi-target regions, and (**D**) ShuEng inference for single-target regions.

**Table 1 sensors-22-07053-t001:** Candidate teacher network; top-1%, parameter quantity, calculation amount, and inference time in NVIDIA Jeston Nano.

Network	Top-1%	Parameter Quantity (M)
VGG16	99.94%	134.27
AlexNet	99.74%	57.00
ResNet18	99.54%	23.51

**Table 2 sensors-22-07053-t002:** Lightweight network; top-1% before and after knowledge distillation, the parameters cardinality, the amount of calculation, and the inference time in NVIDIA Jeston Nano.

Network	Top-1%	KD- VGG16 Top-1%	KD- AlexNet Top-1%	KD- ResNet18 Top-1%	Parameter Quantity (M)	Calculated Amount (M)	Inference Time (s)
**Shuffv2_x0_5**	**98.84%**	**98.09%**	**97.79%**	**99.05%**	**0.34**	**2.95**	**0.0783**
Shuffv2_x1_0	99.26%	98.69%	98.84%	99.63%	1.26	11.62	0.0926
Shuffv2_x1_5	99.50%	98.94%	99.45%	99.71%	2.48	24.07	0.0986
Shuffv2_x2_0	99.67%	99.44%	99.59%	99.73%	5.35	47.62	0.1134
Squeezent1_0	97.86%	88.34%	63.18%	97.94%	0.73	41.74	0.0727
Squeezent1_1	91.20%	96.03%	81.81%	95.23%	0.72	16.05	0.0682
GhostNet	97.94%	99.19%	99.04%	98.99%	3.90	14.26	0.0910
CondenseNetv2	95.12%	98.54%	96.73%	98.69%	7.26	169.0	-

**Table 3 sensors-22-07053-t003:** Effects of different threshold segmentation methods on detection rate, false alarm rate, and average inference time.

Method	Number of Candidate Regions	Detection Rate (%)	False Alarm Rate (%)	Average Inference Time
Otsu	1243	93.49%	9.4%	0.1781
Ours	167	96.74%	4.8%	0.0822

**Table 4 sensors-22-07053-t004:** Detection and false alarm rates in different inference modes in multiple scenarios (unit: %).

Model	Detection Rate	False Alarm Rate
ShuffNet	97.15%	4.87%
ShuffEng	97.15%	4.87%

**Table 5 sensors-22-07053-t005:** Comparison of average inference time between single-target regions and multi-target regions.

	Single-Target Region Inference Time (/s)	Multi-Target Region Inference Time (/s)
ShuffNet	0.0788	0.1293
ShuffEng	0.0341	0.0436

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
