# Peer review of "Photoelectric Target Detection Algorithm Based on NVIDIA Jeston Nano"

_sensors, 2022, doi:10.3390/s22187053_

Round 1
Reviewer 1 Report
In this manuscript, Photoelectric Target Detection Algorithm Based on NVIDIA 2 Jeston Nano was proposed. The identification of the target is analyzed with a comparison of ResNet18. Tensor technology is used for fast inference and obtained accurateness of 97.15% for target recognition with a 4.87% of false alarm rate. The performance displays that the proposed model will be more valuable in the deployment of low computing power devices. I would like to recommend accepting this manuscript.
1. Regarding the methodology, as per my review, the manuscript is well written and explained well.
2. The arguments given in the manuscript support the verdict presented by the experiment.
3. The reference is appropriate
4. The figures presented in the paper are acceptable.
Author Response
Thanks to the advice of the reviewer teacher. In order to make this paper more perfect, some misnomer errors have been revised.
Reviewer 2 Report
This article proposed a photoelectric target detection algorithm to improve object detection.The algorithm achieved high accuracy and low false alarm rate in the experiments. Overall, the innovativeness of this work is good. The article is well-structured. The quality of figures and tables in the article is appropriate. However, there are grammatical errors here and there in the article. I would suggest authors do a thorough review of the article, correct grammatical errors, and break extremely long sentences into shorter ones for better readability.
specific comments:
1. Line 113 - 116, please correct the grammatical errors in this sentence.
2. Line 135-137, please correct the grammatical errors in this sentence.
3. Line 227-234, this sentence is too long and barely readable. Please review the logic of this sentence and break it down to shorter sentences.
4. Line 351 - 354, please break down this extremely long sentence into shorter ones for readability.
5. Line 361-362: "- is indicates that the resources occupied by NVIDIA Jeston Nano are 361 too large to run". Please correct the grammatical errors.
6. For table 2, I would suggest highlighting the best scores in the table for better readability.
Author Response
Thanks to the advice of the reviewer. In order to make this paper more perfect, the following amendments are made to the questions raised by the reviewing teacher:
Q1: Line 113 - 116, please correct the grammatical errors in this sentence.
A1: Changed to: The developed algorithm flowchart is illustrated in Figure 3. Specifically, the acquired active and passive images undergo differential processing and thus are converted to differential images. In the latter images, the proposed adaptive threshold segmentation method is applied based on the statistical characteristics of the echo light intensity of the photoelectric target within the differential image.
Q2: Line 135-137, please correct the grammatical errors in this sentence.
A2: Changed to: This strategy eliminates pseudo-targets and provides the basis for subsequent target identification works. The candidate region discrimination part accurately classifies the candidate region obtained from the convolutional neural network.
Q3: Line 227-234, this sentence is too long and barely readable. Please review the logic of this sentence and break it down to shorter sentences.
A3: Changed to: Consider Figure 7 as an example, in the differential image (a), the maximum gray value of the Target A region (Pseudo-target area), the Target B (True-target Area) region, and the whole image are 198, and 199 respectively. Considering the actual situation, the high brightness reflector gray value is greater than the photoelectric target gray value. Thus, we experimentally found that splitting the high and low brightness region gray values based on the entire image’s highest 1% gray value of the first-pixel point is the most appropriate choice.
Q4: Line 351-354, please break down this extremely long sentence into shorter ones for readability.
A4: Changed to: Figure 9 reveals that the developed method has an appealing effect on suppressing background factors and enhances the contrast of photoelectric targets well, providing convenient conditions for the deep learning model to discriminate the target area. This is especially important as it compensates for the algorithmic complexity of embedded platforms with less computational power.
Q5: Line 361-362: "- is indicates that the resources occupied by NVIDIA Jeston Nano are 361 too large to run". Please correct the grammatical errors.
A5: Changed to: - indicates that the resources occupied by NVIDIA Jeston Nano are too large to run.
Q6: For table 2, I would suggest highlighting the best scores in the table for better readability.
A6: The best scores in Table 2 are bolded for better readability. As follows:
Network |
Top-1% |
KD-VGG16 Top-1% |
KD- AlexNet Top-1% |
KD-ResNet18 Top-1% |
Parameter Quantity (M) |
Calculated Amount (M) |
Inference time (s) |
Shuffv2_x0_5 |
98.84% |
98.09% |
97.79% |
99.05% |
0.34 |
2.95 |
0.0783 |
Shuffv2_x1_0 |
99.26% |
98.69% |
98.84% |
99.63% |
1.26 |
11.62 |
0.0926 |
Shuffv2_x1_5 |
99.50% |
98.94% |
99.45% |
99.71% |
2.48 |
24.07 |
0.0986 |
Shuffv2_x2_0 |
99.67% |
99.44% |
99.59% |
99.73% |
5.35 |
47.62 |
0.1134 |
Squeezent1_0 |
97.86% |
88.34% |
63.18% |
97.94% |
0.73 |
41.74 |
0.0727 |
Squeezent1_1 |
91.20% |
96.03% |
81.81% |
95.23% |
0.72 |
16.05 |
0.0682 |
GhostNet |
97.94% |
99.19% |
99.04% |
98.99% |
3.90 |
14.26 |
0.0910 |
CondenseNetv2 |
95.12% |
98.54% |
96.73% |
98.69% |
7.26 |
169.0 |
- |

Reviewer 3 Report
This author proposes and validates an adaptive threshold segmentation method based on the statistical properties of the optical intensity of the optical target echoes. The knowledge distillation and TensorRT technique are used to improve the accuracy of lightweight networks and accelerate the inference and deployment on hardware platforms, effectively improving the detection of target regions. However, the paper still has some shortcomings, and the following are specific comments.
1. The result in line 17 of the article abstract " The detection rate of this method can reach 29 frames" is not reflected in the later text.
2. The contents of Equation 10 are not explained.
3. In the title of the table in line 476 of the article, Table 4 should be Table 5.
4. The practical application value of this study is duly added.
5. There are grammatical and editorial issues in this paper. Many of them could cause difficulties to the readers.
6. The chart format of this paper is not standardized. Some figures should not have solid lines on the outer ring; The typography of the figures is not aesthetically pleasing; The formatting of the tables in the article should be consistent.
7. In line 476, the table's serial number is mislabeled; Table 4 should be Table 5.
8. In line 227, 7(c) appears in the article, but there is no indication in Figure 7 which figure (c) is the one.
9. The author is suggested to supplement the significance of this research in practical application in the abstract and conclusion.
10. Line 88 mentions that this method ensures the smooth acquisition of active and passive images. Please give a more specific explanation.
11. There is a word case error on line 315.
12. It is suggested that the author make a specific explanation to formula 10.
13. The author is requested to make a more detailed discussion on the results of the comparative test according to Fig. 9, so as to illustrate that the method proposed in this paper has certain advantages.
In summary, it is recommended that the author modify the article.

Author Response
Thanks to the advice of the reviewer. In order to make this paper more perfect, the following amendments are made to the questions raised by the reviewing teacher:
Q1: The result in line 17 of the article abstract " The detection rate of this method can reach 29 frames" is not reflected in the later text.
A1: Add the following to the conclusion: Its running time can be controlled in the range of 0.03-0.05s, and the detection rate can reach 29 frames per second, meeting the real-time processing requirement of at least 20 frames per second in the video detection process.
Q2: The contents of Equation 10 are not explained.
A2: Equation 10 is a supplement to Equation 9, and is explained in the paper. As follows: Equation 10 is a supplement to Equation 9.
Q3: In the title of the table in line 476 of the article, Table 4 should be Table 5.
A3: In the title of the table in line 476 of the article, Table 4 has been replaced by table 5. As follows:
Table 5. Comparison of average inference time between single-target regions and multi-target regions.
|
Single-target Region Inference Time (/s) |
Multi-target Region Inference Time (/s) |
ShuffNet |
0.0788 |
0.1293 |
ShuffEng |
0.0341 |
0.0436 |
Q4: The practical application value of this study is duly added.
A4: Add the following to the conclusion: The proposed method has higher accuracy and lower requirements on hardware computing power and can quickly identify multiple photoelectric targets. The entire process has good real-time performance, making the lightweight network more valuable in deploying low computing power devices.
Q5: There are grammatical and editorial issues in this paper. Many of them could cause difficulties to the readers.
A5: I have carefully reviewed all the information in this article. Many of the errors in this article have been corrected to ensure a better reading experience for the reader.
For example, the first paragraph of the original text: Pinhole cameras, miniature cameras, and other photoelectric equipment popularity bring a severe risk of information leakage, information leakage not only to the society and business brings huge property losses but also severe threats to the personal safety of individuals, for this phenomenon can be used the camera existing cat's eye effect [1] for detection, that is the incident light irradiation camera after the original way back and reflected light intensity than diffuse reflected light intensity 2-4 orders of magnitude higher. The spot generated by the photoelectric sensor or slicing board of the focal plane inside the camera due to the reflection of the cat-eye effect is called the cat-eye target. This method does not rely on the start of the camera and the transmission of wireless signals, only on the physical characteristics of the camera itself, in this field the use of laser-active imaging for detection [2] is one of the most effective ways, mainly used in the detection of conventional cameras.
The first paragraph of the article after revision: Pinhole cameras, miniature cameras, and other photoelectric equipment impose a severe information leakage risk, which may impact society and business, bringing substantial property losses and severe threats to personal safety. Therefore, the camera’s cat eye effect [1] is used for detection, utilizing the incident light irradiation camera after the original way back and the reflected light intensity of the diffuse reflected light, which has a 2-4 orders of magnitude higher intensity. The spot generated on the photoelectric sensor or slicing board of the focal plane inside the camera due to the reflection of the cat-eye effect is called the cat-eye target. This method is independent of the camera’s operation and the transmission of wireless signals and only depends on the camera’s physical characteristics. Thus, using laser-active imaging [2] is one of the most effective ways to detect conventional cameras.
Q6: The chart format of this paper is not standardized. Some figures should not have solid lines on the outer ring; The typography of the figures is not aesthetically pleasing; The formatting of the tables in the article should be consistent.
A6: The chart format for this article has been standardized. The solid lines of some of the outer rings of the picture have been removed and the font has been unified.
Q7: In line 476, the table's serial number is mislabeled; Table 4 should be Table 5.
A7: In the title of the table in line 476 of the article, Table 4 has been replaced by table 5. As follows:
Table 5. Comparison of average inference time between single-target regions and multi-target regions.
|
Single-target Region Inference Time (/s) |
Multi-target Region Inference Time (/s) |
ShuffNet |
0.0788 |
0.1293 |
ShuffEng |
0.0341 |
0.0436 |
Q8: In line 227, 7(c) appears in the article, but there is no indication in Figure 7 which figure (c) is the one.
A8: The content in the diagram is accurately labeled. 7(c) has been replaced by 6(c).
Q9: The author is suggested to supplement the significance of this research in practical application in the abstract and conclusion.
A9: Add the following to the conclusion: The experimental results demonstrate that the developed adaptive threshold segmentation method based on the statistical characteristics of optical target light intensity improves detecting the target region more effectively than the other threshold segmentation methods. The layer-to-layer guidance of knowledge distillation is used to improve the accuracy of the lightweight network and TensorRT technology is used to realize accelerated reasoning on the NVIDIA Jeston Nano hardware platform. The proposed method has higher accuracy and lower requirements on hardware computing power and can quickly identify multiple photoelectric targets. The entire process has good real-time performance, making the lightweight network more valuable in deploying low computing power devices.
Q10: Line 88 mentions that this method ensures the smooth acquisition of active and passive images. Please give a more specific explanation.
A10: Add the following to section 2.1: During image acquisition, the laser and detector are controlled by synchronous triggers. The detector acquires images every time the laser changes its operating state, and the detector’s operating frequency is twice the laser’s, ensuring smooth acquisition of active and passive images.
Q11: There is a word case error on line 315.
A11: The error has been modified. (, The output layer) to (, the output layer).
Q12: It is suggested that the author make a specific explanation to formula 10.
A12: Equation 10 is a supplement to Equation 9, and is explained in the paper. As follows: Equation 10 is a supplement to Equation 9.
Q13: The author is requested to make a more detailed discussion on the results of the comparative test according to Fig. 9, so as to illustrate that the method proposed in this paper has certain advantages.
A13: Add the following: Figure 9 reveals that the developed method has an appealing effect on suppressing background factors and enhances the contrast of photoelectric targets well, providing convenient conditions for the deep learning model to discriminate the target area. This is especially important as it compensates for the algorithmic complexity of embedded platforms with less computational power.

Reviewer 4 Report
Here are some suggestions to improve the quality of the paper from my point of view:
1- The authors should provide a clear comparison study between this work and the recent existing work. In Section 4, you mentioned that "you compared the proposed method with the traditional one". Therefore, readers will have the following logical questions: What do you mean by the traditional method? Why only traditional method not more advanced detection methods using the power of deep learning and AI?
2- The quality of the paper presentation also makes the paper less suitable for a decent journal. Moreover, there are too many abbreviations and typos in this paper, and that makes it harder to read and understand for the readers. The authors should double check the paper. A table of abbreviations will be helpful.
Author Response
Thanks to the advice of the reviewer. In order to make this paper more perfect, the following amendments are made to the questions raised by the reviewing teacher:
Q1: The authors should provide a clear comparison study between this work and the recent existing work. In Section 4, you mentioned that "you compared the proposed method with the traditional one". Therefore, readers will have the following logical questions: What do you mean by the traditional method? Why only traditional method not more advanced detection methods using the power of deep learning and AI?
A1: This article mentions comparison with the traditional method in Section 4.1 and has now been changed to compare with other threshold segmentation methods. As following: Figures 9 (a1) and (b1) illustrate the active and passive images of a single photoelectric target with false targets, while Figures 9 (a2) and (b2) depict the active and passive images of multiple photoelectric targets in a simple environment. The real photoelectric targets are marked with red boxes in the active image. The experimental comparison reveals that the proposed method has more advantages than other threshold segmentation methods for photoelectric target segmentation. In a simple environment, the segmentation effect is ideal, while when there are false targets such as high brightness reflectors, only a part of the false targets can be screened, and the remaining must be further discriminated. In Section 4.2 and Section 4.4, it is mainly a comparison with other deep learning and AI detection methods. As following: Also, the prediction accuracy has a more significant impact on the results. Thus, this index is also one of the factors to be considered. Shuffv2_x0_5 has a higher prediction accuracy of nearly 5% compared to Squeezent1_0 and has a better detection performance. Therefore, choosing the Shuffv2_x0_5 lightweight network is more appropriate to deploy on the hardware platform.
Q2: The quality of the paper presentation also makes the paper less suitable for a decent journal. Moreover, there are too many abbreviations and typos in this paper, and that makes it harder to read and understand for the readers. The authors should double check the paper. A table of abbreviations will be helpful.
A2: I have carefully reviewed all the information in this paper. Correction of many abbreviations and spelling errors in this article makes it easier for readers to read.
For example, the first paragraph of the original text: Pinhole cameras, miniature cameras, and other photoelectric equipment popularity bring a severe risk of information leakage, information leakage not only to the society and business brings huge property losses but also severe threats to the personal safety of individuals, for this phenomenon can be used the camera existing cat's eye effect[1] for detection, that is the incident light irradiation camera after the original way back and reflected light intensity than diffuse reflected light intensity 2-4 orders of magnitude higher. The spot generated by the photoelectric sensor or slicing board of the focal plane inside the camera due to the reflection of the cat-eye effect is called the cat-eye target. This method does not rely on the start of the camera and the transmission of wireless signals, only on the physical characteristics of the camera itself, in this field the use of laser-active imaging for detection[2] is one of the most effective ways, mainly used in the detection of conventional cameras.
The first paragraph of the article after revision: Pinhole cameras, miniature cameras, and other photoelectric equipment impose a severe information leakage risk, which may impact society and business, bringing substantial property losses and severe threats to personal safety. Therefore, the camera’s cat eye effect [1] is used for detection, utilizing the incident light irradiation camera after the original way back and the reflected light intensity of the diffuse reflected light, which has a 2-4 orders of magnitude higher intensity. The spot generated on the photoelectric sensor or slicing board of the focal plane inside the camera due to the reflection of the cat-eye effect is called the cat-eye target. This method is independent of the camera’s operation and the transmission of wireless signals and only depends on the camera’s physical characteristics. Thus, using laser-active imaging [2] is one of the most effective ways to detect conventional cameras.

Round 2
Reviewer 3 Report
The author has incorporated all suggestions in the paper. Now the paper has following observations;
Originality of paper is now looking good.
Technical merit of paper is now looking good.
Applicability of research concept is very good for supply chain management system.
Presentation and English is looking good and have connectivity among all sections.
Overall paper is good and my recommendation to editor that paper will accept as per the journal guideline.
Reviewer 4 Report
The revised version is good and can be accepted for publication